# *Chlamydiae* from Down Under: The Curious Cases of Chlamydial Infections in Australia

**DOI:** 10.3390/microorganisms7120602

**Published:** 2019-11-22

**Authors:** Martina Jelocnik

**Affiliations:** Genecology Research Centre, University of the Sunshine Coast, Sippy Downs 4557, Australia; mjelocni@usc.edu.au

**Keywords:** chlamydial infections, Australia, *Chlamydia trachomatis*, *Chlamydia pecorum*, koala, *Chlamydia psittaci*, avian infections, zoonoses

## Abstract

In Australia, the most researched and perhaps the most successful chlamydial species are the human pathogen *Chlamydia trachomatis*, animal pathogens *Chlamydia pecorum* and *Chlamydia psittaci*. *C. trachomatis* remains the leading cause of sexually transmitted infections in Australians and trachoma in Australian Indigenous populations. *C. pecorum* is globally recognised as the infamous koala and widespread livestock pathogen, whilst the avian *C. psittaci* is emerging as a horse pathogen posing zoonotic risks to humans. Certainly not innocuous, the human infections with *Chlamydia pneumoniae* seem to be less prevalent that other human chlamydial pathogens (namely *C. trachomatis*). Interestingly, the complete host range for *C. pecorum* and *C. psittaci* remains unknown, and infections by other chlamydial organisms in Australian domesticated and wildlife animals are understudied. Considering that chlamydial organisms can be encountered by either host at the human/animal interface, I review the most recent findings of chlamydial organisms infecting Australians, domesticated animals and native wildlife. Furthermore, I also provide commentary from leading Australian *Chlamydia* experts on challenges and future directions in the *Chlamydia* research field.

## 1. Introduction

The diverse members of the phylum Chlamydiae constantly remain in the global spotlight, attracting attention not just as successful and enigmatic human and animal pathogens [1], but also by cutting-edge biological and “omics” research involving Chlamydiae [2,3]. From a global public perspective, Chlamydiae in Australia are perhaps best known as the notorious pathogens of the iconic marsupial, koala (*Phascolarctos cinereus*). Besides the devasting effects on koalas, chlamydial infections more broadly in Australia: remain a significant public health burden; contribute to economic and production losses in the livestock industry; pose a threat to native wildlife health; and pose zoonotic risks to humans. I provide an overview of new and important findings and challenges faced in the process of dissecting chlamydial infections in the Australian setting.

### 1.1. Chlamydia trachomatis: An Australian Public Health Challenge

Since the establishment of the Australian National Notifiable Diseases Surveillance System (ANNDSS) in 1990, *Chlamydia trachomatis* (Ct) remains the cause of the most frequently notified sexually transmissible infection (STI) and a major cause of trachoma in Indigenous (Aboriginal and Torres Strait) populations [4,5,6].

#### 1.1.1. Ct still a Major Cause of Trachoma in a High-Income Developed Country

Trachoma, a preventable infectious cause of blindness, remains endemic in many African, Central and South American, Asian and Middle Eastern developing countries as well as remote regions of Australia, a high-income developed country [7,8]. According to the World Health Organisation (WHO), in 2019 eight countries—Cambodia, Islamic Republic of Iran, Lao People’s Democratic Republic, Ghana, Mexico, Morocco, Nepal and Oman have been validated by as having eliminated trachoma as a public health problem [9]. Although trachoma has been eliminated from wider urban and regional Australian communities since the 1920–1930s [10], Australia has not achieved entry onto the above WHO list. To date, trachoma still remains a public health burden and significant health threat for many remote Indigenous Aboriginal communities, despite many elimination initiatives. Trachoma occurs primarily in remote and very remote Aboriginal communities in the Northern Territory (NT), South Australia (SA) and Western Australia (WA), however cases have also been noted in other states [6,11,12,13].

As elsewhere, trachoma typically manifests as a graded chronic keratoconjunctivitis caused by repeated childhood ocular infections. The repeated infections may lead to scarring with contraction and distortion of the eyelid, which may in turn cause eyelashes to rub against the eyeball (trichiasis), which if left untreated can further lead to blindness [7,13]. The infections can be spread through direct contact of eye discharges, hand-to-eye contact, via fomites or by vectors, and are usually associated with crowded living and poor hygiene and housing conditions [10,13]. As per the national 2017 Trachoma Surveillance report, overall active (inflammatory) trachoma prevalence in young children (up to 9 years) has declined to less than 4% in Australia. Scarring and blinding sequelae can also occur among adult Aboriginal people across the country, however are less prevalent. Interestingly, it was also found that whilst there were more communities with endemic and hyperendemic levels of trachoma, the number of at-risk Aboriginal communities has declined from previous years [14]. This observed decline in active trachoma prevalence and number of at-risk communities can be attributed to several health and hygiene promotion programs throughout the country. A sustained and continued effort in health promotion programs that focus on improving hygiene-related practices (such as clean faces in children) and on environmental health and community improvements (such as improved sanitation and waste disposal) will be crucial in achieving the goal of trachoma elimination in Australia [6,11,14,15].

#### 1.1.2. Sexually Transmitted Ct Infections–STI Infections with No Intention of Slowing Down

In 2017, there were an estimated 255,228 new sexually transmitted Ct infections in Australians aged 15–29 years, and it is postulated that further infections remain undiagnosed or not reported. There were 416.8 notifications per 100,000 people, with notification rates highest in the 20–24 years age group. In the Aboriginal and Torres Strait Islander population, Ct notification rates per 100,000 people were three times as high as those for the non-Indigenous population, with the highest notification rates in the 15–19 years age group [16]. The prevalence of Ct STIs in Aboriginal Australian communities across Australia remains concerningly high [17,18,19]. Overall, the rate of Ct notifications in Australia has increased by 13% in the past two years, with a similar trend seen in both males and females [16]. This trend is not surprising and is congruent with the globally rising trend in overall incidence and prevalence of chlamydial infections, as recently estimated by the WHO team [20]. This report also indicated that the prevalence of chlamydia was highest in the upper middle-income countries, territories and areas.

Sexually transmitted Ct infections affect a myriad of Australian sub-populations (e.g., lesbian, gay, bisexual, heterosexual, men who have sex with men (MSM) and other (pansexual, transgender, queer/questioning, intersex)) and can present from mainly asymptomatic to acute genitourinary, pharyngeal and/or anorectal infection. These symptoms can lead to severe disease sequalae and complications, such as ulcerative proctitis in men and pelvic inflammatory disease (PID) and infertility in women. Furthermore, Ct re-infection is also common as is the risk of increased susceptibility to other infections [21,22,23,24]. As Ct can ascend to the upper genital tract, serious sequalae including PID, which can lead to ectopic pregnancy (EP) and chronic pelvic pain or infertility, are commonly observed in the Australian hospital settings. Data from the past decade (2009–2014) shows increasing Ct-related PID rates [25]. A recent study examining Ct presence in testicular biopsies showed that replicating Ct is present in human testicular tissue in up to 45% detection rates and may be associated with moderate to severe spermatogenic impairment [26]. Thus, the role of Ct in infertility should be considered as a significant factor in both female and male Australians.

Outbreaks of lymphogranuloma venereum (LGV) leading to ulcerative proctitis have persisted in Australia for over a decade among MSM populations [27,28,29,30]. Whilst most of these infections are of an anorectal nature, recent cases of LGV had a clinical presentation of penile ulcers [30]. In general, recent data on prevalence of anorectal Ct infections indicates that these are common, particularly in MSM populations, with diagnosis rates continuing to rise and symptoms diversifying. This is of an increasing public health concern as use of pre-exposure prophylaxis (PrEP) against HIV becomes commonplace in this subpopulation [27,29]. Other subpopulations are also at higher risk than others—data indicated that anorectal Ct infections are present in women and may be more common than anorectal *Neisseria gonorrhoeae* infections [31], with certain populations such as female sex workers at higher risk [32]. Recent study on oropharyngeal chlamydia in MSM also demonstrated that chlamydial DNA can be commonly detected in saliva in a range of 446 copies/mL loads [33], however the epidemiological implications are unknown at present.

Besides the current recommended first-line treatment (a single 1g dose of azithromycin or seven days of 100 mg twice daily doxycycline) for uncomplicated UGT infections [34,35], education, testing, partner notification and re-testing are considered key strategies for chlamydia control [36,37]. Despite the best efforts to control Ct infections in Australia, knowledge gaps remain in the diagnosis, re-testing, and management of Ct infections; not just among the public but also practitioners. For example, adherence to treatment and hence being susceptible to re-infection remains an educational and public health issue. This further highlights the need for continued innovative strategies to improve testing and follow up, particularly targeting sexually active adolescents and young adults, and to maintain education and prevention programs to abate the spread of STIs in Australia [36,38].

#### 1.1.3. The Curious Molecular Epidemiology of Ocular and Sexually Transmitted Ct Infections

In contrast to the abundance of global molecular and local epidemiological data, more molecular studies are certainly needed to dissect the epidemiology of Ct infections in Australia. At present, there is a paucity of molecular data for Australian trachoma Ct strains, further hindering our understanding of epidemiology of this infection. Recent molecular studies have curiously showed that Australian Ct strains associated with trachoma are distinct from other global strains, i.e., those primarily associated with the urogenital tract (UGT) but also classical ocular infections [39]. Whole-genome phylogenetic analyses of ocular Ct isolates obtained from young children with clinical signs of trachoma from a trachoma-endemic region of northern Australia, placed these isolates into two lineages that fall outside the classical trachoma lineage. Instead, the unique and genetically distinct strains clustered within lineages that were previously occupied exclusively by UGT isolates (namely genotypes D–K) [39]. Additionally, both *omp*A genotyping [40] and novel Ct molecular genotyping (CtGEM; developed as a method for high throughput surveillance of strains and the major phylogenetic lineages [41]) of UGT samples obtained from remote indigenous communities also confirmed that Australian ocular strains are distinct from global and Australian UGT strains [40](). Such clustering indicates the polyphyletic evolution of Australian isolates and certainly raises questions around the origins of Ct strains in Australia [39,42], which is an area of continued research.

Regarding molecular characterisation of sexually transmitted Ct strains, the knowledge gap is even more prominent. Globally, the emergence of new sexually transmitted strains and variants and their rapid spread indicate the need for more comprehensive molecular studies to better understand the Ct tissue tropism as well as epidemiological network structures [43,44,45]. Molecular assays targeting the *omp*A gene revealed that Ct genovars E and F are commonly found in Australian heterosexual men and women, while genovars G and D are the most common in MSM with genovar L2 also occurring [46]. Recent multilocus sequence typing (MLST) of chlamydia-positive rectal samples from MSM and self-collected UGT swabs from heterosexual women and men from a cosmopolitan setting revealed that most infections reflect the global diversity—sequence types (STs) associated with MSM (ST52, ST58, ST108, ST109) and heterosexuals (ST3, ST12, ST55, ST56) were delineated [23]. In the same study, seven new STs not described anywhere else in the world were also characterised, five identified within MSM and two within heterosexuals, indicating that the epidemiology might be more complex than previously thought [23]. Molecular Ct studies can certainly be improved in Australia as they can be utilized as a valuable epidemiological tool to track new and/or persistent infections or determine treatment failure, particularly as there is growing concern about the latter [23,41,46].

### 1.2. Chlamydia pneumoniae—An Elusive Human Cosmopolitan Pathogen

Globally, *Chlamydia pneumoniae* (Cpn) is reported in a spectrum of diseases involving many body systems [47,48]. In Australia, Cpn infections are mostly associated with the respiratory tract however infections associated with complications post lung transplants, cardiovascular disease and atherosclerosis, and even age-related macular degeneration (AMD) were also noted. Certainly not innocuous, Cpn seems however to be less prevalent that other related human chlamydial pathogens.

Perhaps a contributing factor to the under-detection of Cpn is that Cpn is not considered a major etiological agent in cases of community or hospital acquired pneumonia or other severe respiratory disease [49]. Recent studies somewhat confirm these observations. The Childhood Infectious Diseases study conducted in Queensland, Australia from September 2010 through October 2014, followed 158 young children from birth until their second birthday and tested 8100 high-quality nasal swab specimens for respiratory bacteria, including Cpn [50]. Cpn was identified in only two swabs. Similarly, Rane and colleagues detected Cpn in nasopharyngeal swabs or aspirate samples from only three patients (two children and one adult) presenting with respiratory symptoms, and only after broadening the detection method to include the family Chlamydiaceae [51]. Interestingly, the same study also identified for *C. trachomatis* in four children under the age of 2 years, and *C. psittaci* in three adult patients, also presenting with respiratory symptoms. In all these cases, *Chlamydia* spp. infections had not been anticipated by the requesting clinician, further indicating that these are likely to be under-recognized causes of serious respiratory tract infections [51]. In another study estimating point prevalence for viruses and bacteria in 244 children aged 2–16 years presenting with acute asthma, Cpn was not detected in any of the nasopharyngeal aspirate samples [52]. In contrast, a study from the early 2000s examining the relationship between airway inflammation and serological response to Cpn in acute severe asthma in adult patients aged 16–74 years, with the majority having at least moderatly persistent asthma, found that >30% of adults presenting with acute severe asthma showed a rise in Cpn-specific antibodies consistent with acute infection, reinfection or reactivation of latent infection with Cpn [53]. The limitation of such studies is that Cpn infection had been inferred serologically without isolation or more sensitive molecular detection of the organism.

The pathogenic role and detrimental impact of Cpn was demonstrated in cases of Cpn infections post lung transplantation [54]. In this study, bronchoalveolar lavage (BAL) samples were taken from 80 lung transplant recipients during 1-year post transplantation and PCR screened for Cpn. Nine recipients were positive for Cpn within 30 days of lung transplantation, three of whom remained positive on repeat lavage and died from airway disease in the first year post-operatively. A number of other patients were positive for Cpn at >30 days after lung transplantation, and developed serious sequalae such as pneumonia, bronchiolitis obliterans syndrome and acute pulmonary allograft rejection [54]. Persistent infection with Cpn in organ transplants surgeries (whether donor-derived, de novo or re-activated) represents serious health risk.

The link between Cpn and cardiovascular disease is globally well established [55], however association of Cpn with atherosclerotic plaques in Australian subjects has been somewhat contradictory and under debate [56,57,58,59]. A study from the year 2000 examined 33 atherosclerotic coronary artery segment specimens, 13 diseased vein graft specimens and 10 segments of internal mammary artery did not find Cpn either by PCR or culture, despite >20% seroposivity [58]. However, a subsequent study detected Cpn in 15 of the atheromatous lesions as well as in three of the normal tissues [57], further suggesting that Cpn infection might be an initial trigger of atherosclerotic development. Cpn infections have also been reported in adult patients with multiple sclerosis, namely in venous obstructions due to Cpn-associated chronic persistent venulitis [60]. An increased risk of Cpn-associated cardiovascular events in HIV-infected Australians have also been evaluated [61], however the role of chronic infections (such as Cpn) in immunosuppressed patients, with respect to the possible increase in vascular events noted in association with HIV infection, remains to be elucidated.

Age-related macular degeneration (AMD) is the leading cause of unavoidable visual loss affecting ageing (over 60 years of age) Australians [62] and the proposed role of and Cpn involvement in AMD [63] has been also investigated in Australian cohort. A study by Robman and colleagues investigating an association between exposure to Cpn infection and progression of AMD in patients aged 51–89 found that the rate of AMD progression over a 7-year period was increased in those with higher Cpn antibody titers [62]. At present, more studies are needed to elucidate the role of Cpn in AMD [63].

Cpn infections are also an important contributing cause of overall high morbidity and mortality in Australian Aboriginal populations [8,64]. Cpn incidence/prevalence remains difficult to quantify in Australia [49,51,65], nevertheless, there is an evidence that Cpn infections are common in Australians within cosmopolitan settings, but the role of Cpn as a pathogen is elusive.

#### Limited Molecular Epidemiology of Australian Cpn Strains Reveals Distinction between Animal and Human Strains

At present, Cpn epidemiology is also hampered by the lack of molecular characterisation of the detected strains infecting Australians. Studies from the 2010s highlighted that animal isolates are much more genetically diverse, yet distinct from human Cpn isolates, with observations that human strains were zoonotically acquired on at least two occasions in the evolutionary history of this pathogen [66,67,68]. A subsequent study using comparative genomics and phylogenetic analyses, demonstrated a distinct Australian indigenous Cpn clade pre-dating European exploration of the continent [64]. The same study also confirmed that modern human Cpn strains have evolved separately from animal strains, further demonstrating the very curious evolution of this species.

## 2. Chlamydial Infections at the Human–Animal Interface

### 2.1. Chlamydia psittaci—A Parrot Pathogen with Zoonotic Potential

*Chlamydia psittaci* (Cps) infections are not new in Australia; in fact, the research on Cps started in the 1930s, almost coinciding with the global psittacosis epidemic and was led by the Nobel Prize winner and renowned Australian virologist and immunologist Sir Frank Macfarlane Burnet [69]. Besides his early work on chlamydial cell biology, in 1934–1935 Macfarlane Burnet demonstrated that Cps infects apparently healthy and wild parrots [70] and wild Australian parrots but could cause disease when parrots were stressed under conditions of confinement by bird dealers [71]. In his follow up studies, he also investigated outbreaks of fatal psittacosis in wild parrots [72].

Human Cps infections are notifiable in Australia, however are very rare, with only approximately 50 notified cases of psittacosis per annum in the past decade according to the ANNDSS. As Australia is home to many wild bird species, direct as well as indirect contact with birds is well established and is considered a common risk factor for the human psittacosis [73,74,75,76,77]. Considering the wide spectrum of clinical manifestations for psittacosis (asymptomatic to mild respiratory to fulminant systemic disease), asymptomatic and/or mild cases could be easily missed, misdiagnosed or self-resolved, and as such not notified. As noted above, the Cps infection rate in Australia is probably higher than diagnosed [51]. Globally, Cps is a) an established significant zoonotic pathogen with focal outbreaks commonly reported [78] and b) estimated to be cause in approximately 1% of annual community acquired pneumonia (CAP) [79]. Due to this, effective management of Cps outbreaks should be incorporating and implementing the “One Health” approach [80,81].

#### 2.1.1. Equine and Avian Psittacosis: On the Heels of Cps Infection “Spill-Over”

Whilst many of the described human psittacosis cases involve birds as a reservoir or zoonotic factor, in 2014–2017, cases of equine reproductive loss came under the spotlight due to apparent zoonotic transmission of Cps from equine placental membranes to humans, a previously unrecognised route of transmission for this organism [81,82,83]. Curiously, the surprise was not the zoonotic event itself, but rather the detection of Cps and associated pathology in “novel” hosts, the Australian thoroughbred horses. Screening for Cps in horses was prompted by a 2014 index case of mare abortion from the largest thoroughbred region in regional New South Wales (NSW), Australia, and consequent zoonotic transmission of Cps from the aborted material to humans resulting in five cases of psittacosis [82,84]. During the 2016–2017 foaling season, a Cps-associated epizootic of horse reproductive loss occurred in the same region. The diagnostic work-up of foetal and placental tissue samples from cases of equine abortion and foals with compromised health status revealed Cps positivity of up to 23%, with detection of high chlamydial loads in samples [83]. Cps was also described in cases of acute respiratory distress in neonatal foals [85], bringing into question the tissue specificity of Cps in horses and the mode of transmission between horses. Soon after the NSW epizootic, another case of Cps-associated equine abortion in the geographically separated Queensland (Qld) region of Australia was described [86]. These findings together suggest that a) Cps may be a more significant pathogen of Australian horses than previously thought, b) equine chlamydiosis may have resulted from spill-over of infected birds and c) Cps should be considered a differential diagnosis for neonatal foals and pregnancy losses. While it was postulated that equine Cps infections could be a result of Cps spill-over from birds (Figure 1A), none of the studies assessed sympatric birds for Cps shedding, which should be a focus of future work.

Detection of Cps in grazing livestock (cattle, sheep, horses and pigs) is common globally, and it is estimated that Cps infection/detection rates can be as high as 50% in a herd while the pathogen itself causes a range of diseases (such as ocular infections, pneumonia, vaginitis and mastitis) [1,78,87]. In contrast, until recently the reports in Australia were anecdotal, inconclusive and rare. Since the description of Cps infections in horses, Cps has been detected in a limited number of domesticated cattle and sheep [88,89], however wider surveys are warranted to truly elucidate whether the livestock animals are indeed commonly infected with this pathogen and represent “new” reservoirs, or if Cps detection is a result of “spill-over” from birds and/or environmental exposure (Figure 1A).

In the recent reports, Cps infection in birds was usually reported as a part of (a) the zoonotic event, such as that of molecular characterisation of Cps isolated from a crimson rosella from the same human psittacosis endemic region [90] or (b) horse psittacosis investigations, such as that of an opportunistically characterised archival Cps-positive sample taken from a spotted dove [86]. Both perplexing and in contrast to the 1930s studies, detailed studies about Cps prevalence in Australian domesticated/pet or wild birds are lacking. A recent opportunistic survey of more than 400 wild and captive birds presented for veterinary care during December 2014 and December 2015 to the Wildlife Health Centre in Victoria (Vic) showed a significantly higher prevalence of infection in captive birds (8%; 9/113) compared to wild birds (0.7%; 2/299) [91]. In this study, Cps was characterised from a wild crimson rosella and sulphur-crested cockatoo, captive scaly-breasted lorikeets, cockatiels and a single pet budgerigar [91]. In another study from the same state, Cps was detected in single little corella out of 55 wild cacatuids tested [92]. These studies, albeit small, give an indication of the low prevalence of Cps in the wild and raise questions over the role of birds as wildlife reservoirs in Australia.

#### 2.1.2. Are Australian Birds Really the Culprit in the Cps Infection “Spill-Over”?

In contrast to the above, molecular investigations of the human psittacosis, equine reproductive loss and bird surveillance studies continue to support the “birds as Cps reservoirs” hypothesis. Whole genome analyses of human Cps isolates from an endemic region in NSW were nearly identical to strains detected in the sympatric crimson rosella parrots [90], with all strains tightly clustering in the globally distributed, highly virulent, clonal “parrot” 6BC ST24 clade [93]. Genomic analysis and molecular typing of Cps-positive horse samples from the mare abortion index case and Cps-equine epizootic revealed that the horse strains from different studs in regional NSW were clonal, again clustering tightly with previously described Australian parrot and human isolates within the ST24 clade [83]. Not surprisingly, whole genome analyses of Cps detected in the little corella from Victoria also clustered this strain in the ST24 clade [92].

However, MLST of the Cps horse placental strains from the Queensland abortion case and a dove from NSW revealed that the Cps strains were genetically diverse. Both STs (equine ST27 and dove ST35) were closely related to the strains typically associated with infections of pigeons [86]. MLST analysis on Cps-positive samples from the bird surveillance study from Vic also detected Cps ST24 in both wild and captive birds, while Cps ST27 (detected in horses) was now unexpectedly described for the first time in a wild sulphur-crested cockatoo [91]. The Australian sheep and cattle Cps strains are yet to be molecularly characterised to confirm whether they fit within the “Cps bird reservoir” hypothesis. Nevertheless, limited available data indicates that Australian Cps diversity reflects the global Cps diversity, and we may continue to uncover novel Cps diversity genotypes. In light of recent Cps “spill-over” to a new host, the horse that further “spilt-over” to humans, and continued reports of human and avian psittacosis, broader bird and livestock surveillance/molecular epidemiology studies are imminent (Figure 1A), while appropriate hygiene and biosecurity practices are highly recommended due to the serious human health implications of infection with this pathogen [83,91].

## 3. Chlamydial Infections at the Domesticated–Wildlife Animal Interface

### 3.1. Chlamydia Pecorum—The Infamous Koala or Economically Significant Livestock Pathogen?

Cpec remains the major pathogen of the Australian iconic marsupial, the koala, posing a very serious threat to the long-term survival and conservation of this unique native species and the only extant member of the family Phascolarctidae [94,95]. Koala chlamydiosis is certainly one of the most researched wildlife diseases in Australia, with observations that Cpec infects and causes disease in koalas noted since the early 1900s [96]. As such, koala chlamydiosis also serves as a wildlife disease model, and many lessons from koala chlamydiosis can be transferred and utilised in other research. In contrast, the observations and impact of ubiquitous and endemic Cpec infections in Australian livestock only started to gain recognition in the last decade [97], and in this case Australian research taking notes from global livestock studies.

#### 3.1.1. Cpec Infections in Koalas—(Still) a Major Cause of Debilitating Disease

Koala Cpec infections and the resultant debilitating disease deserve rather constant updates to aid in an ongoing tireless effort in the control of these infections and successful management of the infected koalas. Cpec infects almost all of Australia’s mainland koala populations spanning from northern (Qld and NSW) to the southern states (Vic and South Australia (SA)) [98,99,100,101,102]. However, a recent study estimating Cpec prevalence in SA koala populations, showed that the geographically separated Kangaroo Island koala population was infection- and disease-free, whereas the mainland population was not [103]. These koalas could provide a safeguard against this serious disease threatening to decimate this iconic Australian species [103]. Cpec is readily detected at the ocular, nasal, urogenital and rectal sites in a range of infectious loads, infecting both female and male young joeys to mature adult koalas [99,100,104,105,106]. If infection progresses to clinical disease, the classical manifestations are inflammatory and fibrotic disease in the eye and the urogenital and reproductive tracts [107,108]. Recently, the spectrum of disease caused by Cpec in koalas was also extended with the descriptions of a syndrome of rhinitis/pneumonia [109] and arthritis affecting one or more joints [110]. Reports suggest that perhaps there is a difference in chlamydial disease severity between northern and southern koala populations, with a higher prevalence and disease severity readily reported in the northern koala populations [100].

It is interesting to note that traditionally, chlamydial disease is studied without consideration of other co-infections [111]. However, as the immunosuppressive koala retrovirus (KoRV) is also ubiquitous in these animals, the role of KoRV co-infections in chlamydial disease progression in koalas must be acknowledged [112]. There are two major subtypes of KoRV infecting koalas: KoRV-A, believed to be endogenous only in koalas from the northern part of Australia however moving south, and KoRV-B, which appears to be exogenous and only present in the northern koalas [112,113]. Out of the two types, studies link KORV-B to neoplasia and chlamydial disease in both wild and captive koalas, making it a double threat to this already vulnerable marsupial [111,112].

Further challenges to treating diseased koalas are a) only two antibiotics are available—chloramphenicol as the drug of choice, and enrofloxacin, b) surgical interventions, such as ovario-hysterectomy, require prolonged recovery time and induce stress to the animal and c) not all koalas are candidates for treatment [107,108]. In most cases, chronically diseased animals, animals in poor body condition, or with comorbidities such as KoRV-associated disease are not suitable candidates as antibiotics can be detrimental for the koala′s gastrointestinal tract microbiota and in severe cases, can lead to dysbiosis and death [99,108]. Novel compounds, such as a serine protease inhibitor (JO146), that inhibits proteolytic activity of the chlamydial putative virulence factor high temperature requirement A (HtrA) [114], and alternative antibiotic therapy with doxycycline and topical supportive therapy have also been proposed [107].

Hence, more than ever we turn to a koala Cpec vaccine as a promising infection and disease control strategy. Evidenced by ongoing successful vaccine trials using both recombinant and peptide vaccine formulations in wild and captive koalas, the koala Cpec vaccine development has made tremendous progress in the past decade, as recently reviewed by Phillips et al. [115]. Briefly, (a) the vaccine is safe to use in both healthy and infected koalas, providing level of cross protection against a variety of Cpec MOMP genotypes [116,117,118,119], (b) both humoral and cellular immune responses are elicited and are long-lasting [120] and (c) vaccination has both therapeutic and prophylactic effects [121,122].

Considering such high Cpec prevalence in koalas, Cpec infections in other native non-koala marsupials and wildlife are understandable. Since the early 2000′s, Cpec has been described in the greater glider, mountain brushtail possum and western barred bandicoot, with recent reports expanding the host range to the common brushtail possum, squirrel glider and spotted tail quoll [123]. The exact impact of Cpec infection in these hosts is however unclear and remains open for further investigations. It is interesting to note that to date Cpec (nor other chlamydiae) was not described in the kangaroos nor wombats, other ubiquitous marsupials.

#### 3.1.2. Cpec Infections in Livestock—A Contributor to Economic and Production Losses

Globally published data suggests that Cpec is essentially ubiquitous in livestock (sheep, cattle, goats and pigs) [124]. Unfortunately, the understanding of the epidemiology of livestock Cpec infections in Australia only started to increase in the past decade. One of the main drivers to investigate epidemiology of Cpec in livestock were questions over the potential role of cross-host transmission in the origin and epidemiology of chlamydial disease in koala, considering that the encroachment of koala habitats by sheep and cattle farming along the east coast of Australia is common [96]. At present, most of the studies have focused on the agriculturally productive region in central NSW, where endemic Cpec infections have been described in sheep and beef cattle. In these hosts, Cpec is mainly detected as asymptomatic shedding in mature animals, and manifests more acutely as keratoconjunctivitis, polyarthritis, conjunctivitis with polyarthritis and Sporadic Bovine Encephalomyelitis (SBE) in young animals [97,125,126,127]. There are also anecdotal reports of Cpec-associated sheep abortions. However, a study assessing the prevalence via faecal shedding of *Chlamydia* spp. in sheep across Australia estimated that Cpec is prevalent in at least 30% of the nation’s sheep flock [128]. Limited reports support that Cpec infections are common in livestock throughout Australia, and not just contained to geographically distinct regions. Cases of Cpec-associated SBE and shedding have been reported in cattle from Western Australia (WA) [127], whilst our recent study on chlamydial infections in dairy cattle from Qld revealed that Cpec infections are common and could be highly prevalent (up to 50%) in dairy herds [129]. The same study also suggested that Cpec infection may impact dairy herd health at the production level rather than affecting individual animal.

In practice, treatment of Cpec-diseased animals displaying polyarthritis, SBE and conjunctivitis is with the use of intramuscular injections of long-acting oxytetracycline (300 mg/mL at a dose rate of 1 mL per 10 kg bodyweight) once a week, twice [97]. However, the antibiotic treatment is partially effective as evidenced by numerous case relapses [97,130]. As is for koalas, a vaccine for livestock chlamydioses seems like a highly attractive alternative. A recently developed prototype Cpec vaccine, comprised of a major outer membrane protein (MOMP-G) and polymorphic membrane protein G (PmpG), was safe to use in pregnant sheep and lambs, and triggered production of systemic anti-MOMP-G and anti-PmpG IgG antibodies, secretory IgA in the ocular mucosa, and promoted production of IFN-γ in lambs [130].

While the exact economic and production losses incurred due to these infections remains unquantified, Cpec infections pose a problem in the thriving Australian live export industry [131,132]. *Chlamydia abortus* (Cab) is considered an exotic pathogen in Australia and New Zealand, with many recent surveillance studies supporting this observation [89,129,131]. In a recent pilot study, it was shown that commercially available Cab-specific assays demonstrate low specificity, yet high rates of Cab seropositivity in laboratory confirmed Cpec infected Australian sheep and in the absence of any detectable Cab infection [131]. As such, the use of these assays may be contributing to unnecessary rejection of animals in various international markets that require Cab testing, consequently reducing profitability and incurring losses to the industry. A recent study screening for chlamydial pathogens in conjunctival swabs collected from rejected sheep with varying grades of infectious ovine keratoconjunctivitis (IOK) from an Australian pre-export feedlot also demonstrated that Cpec ocular infections are common in both sheep presenting with IOK, as well as healthy animals [89]. These findings further support a role for Cpec infections in economic losses to the Australian sheep industry.

Besides detailed reports and studies of Cpec infections in Australian sheep and cattle, at present it is unknown whether the same pathogen infects broader livestock hosts or not. In a recent survey of wildlife and domesticated ungulates, Cpec was detected in ocular and rectal samples from only three pigs from the same endemic region as the sheep and cattle [88]. Similar to the Cps livestock infections, the status of “true” infections in expanded livestock hosts remains open for further investigations, as Cpec detection could have been due to environmental exposure from sheep and cattle faecal shedding. The lack of studies in this area indicates a role for further sympatric sampling and molecular epidemiology.

#### 3.1.3. The Complex Molecular Epidemiology of Koala and Livestock Infections

Until a more comprehensive set of Cpec whole genome sequences from koalas and Australian livestock (only 5 genomes from each have been published thus far) is available [133,134], Cpec-specific MLST and *omp*A genotyping have proven to be effective molecular epidemiology tools for characterising strains infecting these hosts [98,100,135,136,137,138]. Whilst highlighting an unexpected level of diversity for not just Australian strains but the global Cpec population, these tools have provided the following important epidemiological observations in Australia: (a) a single host can harbour two distinct strains at different anatomical sites, (b) a mixture of genetically diverse strains can circulate in a single population, (c) the same strain can infect two different hosts (e.g., koala and sheep or sheep and cattle(Figure 1B)) and (d) some koala strains are more genetically similar to livestock strains than they are to other koala strains (Figure 1B) [136,137,138]. Molecular genotyping studies also suggested that certain Cpec genotypes may demonstrate tissue tropisms and pathogenicity. For example, in sheep and cattle, the genotype ST23 has been repeatedly observed in both sheep with polyarthritis and conjunctivitis, and cattle with SBE (Figure 1B) [135]. Similarly, in koalas, *omp*A strain type E’ was detected in association with both ocular and UGT disease [100]. Furthermore, preliminary observations also indicated a possible association between Cpec plasmid-bearing strains and clinical disease [105,139].

Both MLST and genome-derived phylogenies highlighted the polyphyletic evolution of Cpec infecting Australian animals and provided important clues about the origins of koala strains [134,139], somewhat supporting the hypothesis that the origin of Cpec infections in Australia is associated with importation of domesticated animals with European colonisation. Even so, we must note that none of the studies sampled sympatric livestock and koala hosts, making it challenging to obtain more concrete evidence on Cpec spill-over from livestock to koalas. Perhaps, comprehensive analyses of whole genomes sequences from the global Cpec population might finally resolve questions about origins of this pathogen in koalas.

## 4. Other Chlamydial Infections in Australian Animals: We Can’t Find What We Don’t Seek

It is almost unimaginable that reports of other chlamydial infections are so scarce in Australia considering (a) global animal and human movement, (b) infections with other chlamydial species are common in overseas animals and (c) we have a very active domesticated animal/wildlife/human interface. This lack of reports may be due to many factors, including the unique geographical positioning of Australia and its strict biosecurity regulations aimed at decreasing risks and and helping Australia remain free of severe pests and diseases [140] and the fact that these infections are asymptomatic and as such do not pose a concern; and research emphasis is placed on the most significant human, zoonotic and wildlife chlamydial pathogens (namely Ctrac, Cps, Cpec and Cab). Still, recent reports of novel or emerging chlamydial infections in Australia have been prompted whilst focusing on the traditional pathogens (Figure 2A).

Besides Cpec and Cps, a recent study detected *Protochlamydia amoebophila* in a cow; novel chlamydial sequences highly related to *C. suis* and *C. trachomatis* as well as novel lineages closely related to *Protochlamydia* species in horses; and uncultured chlamydiae that may represent a novel family-level lineage that lies between the *Parachlamydiaceae* and *Chlamydiaceae* families in free-range deer (Figure 2A) [88]. Similarly, despite its global distribution in avian and livestock hosts, *C. gallinacea* in Australian birds has only recently been identified in cloacal samples from 36 chickens, and in one sample from a female galah with healthy appearance [141]. In both studies, domesticated animal and wildlife habitats overlap making cross-host transmission plausible. Several studies also described emerging chlamydial infections, manifesting as epitheliocystis, in both wild and farm fish across Australia such as wild and farmed Striped trumpeter [142], and more recently farmed Orange-spotted grouper [143]. Regarding other chlamydial pathogens, (a) to the best of our knowledge there are no current reports about tetracycline resistant nor tetracycline sensitive *C. suis* [1] in Australia, (b) the reports about Cpn infections in native animals, namely koalas, have drastically declined and (c) the recent reports about *C. felis* [144] are scarce, whereas *C. caviae* infections remain anecdotal. These findings indicate a plethora of new reservoirs and expanding chlamydial diversity that could be further investigated (Figure 2B).

## 5. Present Challenges and Future Directions

Exciting times are ahead for chlamydial research in Australia and all human and animal chlamydiologists as we embrace challenges with a common goal of successfully managing and controlling these infections. I wish to conclude this review with provided commentary from leading Australian chlamydia experts on challenges and future directions in the chlamydia research field.

When talking about Ct challenges and directions, Professor Jane Hocking said “Chlamydia continues to be the most commonly diagnosed bacterial sexually transmitted infection among young adults with ongoing challenges to reduce its burden in the population. We now know that repeat chlamydia infection in women increases their risk of developing pelvic inflammatory disease (PID), so our research is investigating interventions to reduce the risk of repeat infection among those diagnosed with chlamydia. These interventions include strategies to treat the sexual partners of those diagnosed with chlamydia, novel approaches to increase chlamydia re-testing three months after diagnosis and treatment to detect repeat infections early enough to reduce the development of PID and interventions to improve the early detection of PID in general practice. An important priority for future chlamydia research is to identify biomarkers that can identify chlamydia infections that are likely to ascend the endocervical tract in women causing PID. And of course, research into an effective chlamydia vaccine!”

Regarding chlamydial disease in koalas, Dr. Rosemary Booth said “Chlamydiosis is a key threatening process for koalas. It is primarily a sexually transmitted disease which causes devastating symptoms including blindness, chronic cystitis and infertility. Treatment is possible using the same broad-spectrum antibiotics that are used in people, but because koalas are presented to wildlife hospitals with severe chronic disease, chronic inflammation can persist. Once a koala returns to the wild, it can become re-infected at its next sexual contact. To reduce the impact of Chlamydiosis on koalas, the focus needs to be on retaining connected and protected habitat allowing them to maintain good genetic variability and have good nutrition that supports immune function. While work continues to improve the connectivity of good koala habitat, the development of vaccines against Chlamydia and koala Retrovirus will benefit wild koala populations.” Professor Peter Timms further said “Chlamydial disease continues to be a major threat to the longer-term survival of the koala. While antibiotics are available to treat koala *C. pecorum* infections, this causes serious gut dysbiosis and often death in diseased koalas. A major goal therefore, is the development of a vaccine for *C. pecorum* and research in both captive and wild koalas is showing great promise. Just as important is the development of specific anti-chlamydia antibiotics. Of all chlamydial infections in animals, *C. pecorum* disease in the koala is perhaps the most extreme, and hence understanding the mechanisms of disease pathogenesis is critical.”

Regarding molecular epidemiology studies in Australia, Dr. Alistair Legione said “Expanding detection, typing, and genomic sequencing of *Chlamydia pecorum* into other species, particularly asymptomatic livestock populations that we know are likely to carry the organism (cattle, sheep, pigs), is crucial to us understanding the broader epidemiology of the infection in koalas. In particular tracking the origin of koala infections, whether there is sustained or regular spill-over from source livestock populations or whether transmission happened as a rare event 200 years ago (or if at all), is impossible to answer with the current gaps in the knowledge. Further, our understanding of this pathogen in koalas is restricted to assessing it in populations undergoing rigorous management, or animals that are admitted to wildlife hospitals. This leaves large fragments of the koala population in Australia unaccounted for in regard to prevalence of the pathogen and effect on the koalas. Whilst sampling animals in these wild locations is difficult, new techniques using detection dogs and extraction of chlamydia DNA from faecal samples may help close these gaps at least in respect to prevalence and epidemiology.”

Thoughts about the emerging *C. psittaci* infection in livestock are similar. Future investigations must be planned in more detail to include longitudinal sampling, compulsory sampling of sympatric hosts (i.e., horses and birds or co-grazing livestock from the same studs) as well as healthy animals, co-infections with other significant pathogens, all accompanied with detailed metadata (i.e., farm practices, animal clinical data, location) to truly understand the spill-over event itself as well as the risks of it. We also need to focus and expand on spill-over risks to the public health. Furthermore, Dr. Rosemary Booth also said “Avian chlamydiosis causes chronic respiratory disease in birds and can lead to life threatening illness in compromised birds with poor nutrition or co-infection with other pathogens. It is also a zoonotic disease capable of causing a fatal pneumonia in humans. Wild bird populations are at risk when their natural habitat is fragmented, and when they are concentrated into smaller feeding areas which favour the transmission of air borne pathogens. The main bird species affected are parrots and pigeons, although it can be found in other species as well.” 

Regarding chlamydial cell biology studies done in Australia, Associate Professor Wilhelmina Huston said “Studies to examine chlamydial cell biology are still hampered by the intensive nature of the work to conduct culture and manipulations. Although there have been many advances in genetic manipulation, and other cell biological tools in the chlamydial field, these still require a lot of investment of time and patience. This means it can be hard to justify molecular and cell biology studies of field isolates and new isolates, when much of this work hasn’t been conducted in ‘type’ strains. It may be that the most recent clinical and animal field isolates are actually best to target for new molecular studies, especially looking at animal or tissue niche or unique virulence traits.” 

Regarding chlamydial “omics” studies, Associate Professor Garry Myers noted, “In the 21 years since the first Chlamydia genome was published by Stephens and colleagues in 1998), genome-scale sequencing has been a critical factor for advancing our understanding of chlamydial biology, particularly for defining the breadth of chlamydial and chlamydia-like diversity in animals and the environment. Nevertheless, the obligate intracellular niche shared by all members of the genus and their multiphasic developmental forms continue to render chlamydia difficult to study. This is exemplified by the relative difficulty of chlamydial transformation, despite significant recent advances. The next wave of functional genome-scale analyses is starting to delve into the foundations of disease by deciphering the complex interactions of chlamydia with infected cells, tissues and the mammalian immune response.”

While understanding pathogenesis of disease and genetic makeup of the infecting strains is of great importance, another area of improvement is the development of rapid and point-of-care usable chlamydial diagnostics. The importance (and necessity) of rapid but accurate detection was evident in horse Cp-associated abortions, particularly if zoonotic event was suspected. This also applies to human Ct infections and koala chlamydiosis to efficiently treat and manage infected hosts.

## Figures and Tables

**Figure 1 microorganisms-07-00602-f001:**
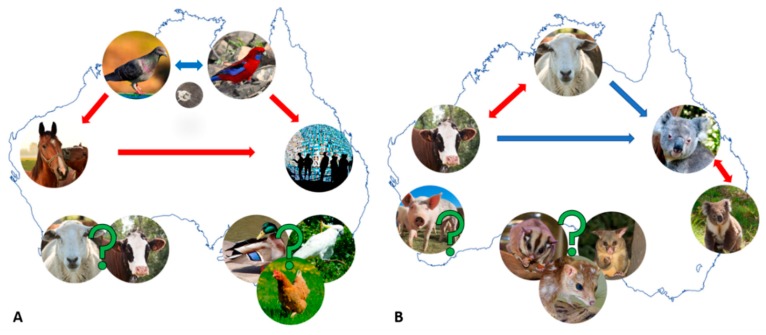
Schematic representation of postulated epidemiology and cross-host transmission of the chlamydial infections in Australia. (**A**) Postulated cross host transmission of *Chlamydia psittaci* (Cps) infections and (**B**) Postulated cross host transmission of *Chlamydia pecorum* (Cpec) infections. Red arrows indicate the most likely cross-host transmission routes supported by molecular investigations (e.g., in **A**, Cps from birds or horses to humans and in **B**, Cpec between koalas or livestock). Blue arrows indicate likely (hypothesised) routes of cross-host transmission (e.g., in **A**, Cps between birds and in **B**, livestock Cpec to koalas) and green question marks indicate scarce reports in the denoted hosts and mainly unknown epidemiology.

**Figure 2 microorganisms-07-00602-f002:**
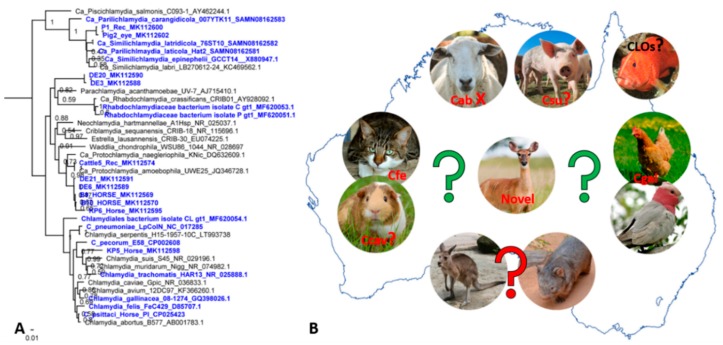
Genetic diversity of chlamydial infections in an expanded host range in Australia. (**A**) a mid-point rooted maximum likelihood phylogenetic tree constructed using an alignment of the 298bp “chlamydial” signature fragment of the 16S rRNA gene from the publicly available sequences from Genbank. Support values are shown on the major nodes. Phylogenetic positioning of taxa described in Australian hosts are highlighted in bold and blue. (**B**) Schematic representation of an expanded range of animal hosts for chlamydial infections, where X denotes exotic chlamydial species (e.g., *Chlamydia*
*abortus*), red question marks indicate no or anecdotal infection or hosts reports (e.g., *Chlamydia*
*caviae* and *Chlamydia*
*suis,* wombats and kangaroos), and green and black (for chlamydia-like organisms (CLO) in fish) question marks indicate scarce reports in the denoted hosts and mainly unknown epidemiology.

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
