# Peer review of "Chlamydiae from Down Under: The Curious Cases of Chlamydial Infections in Australia"

_microorganisms, 2019, doi:10.3390/microorganisms7120602_

Round 1
Reviewer 1 Report
This review expertly summaries the current work and knowledge on the host distribution and epidemiology of Chlamydia species in regards to Australia.
The review briefly summarises the current status of C. trachomatis and C. pneumoniae with a deeper dive on the current knowledge for C. psittaci and C. pecorum.
I believe this review stands apart from other recent Chlamydia reviews by focusing on broad range of pathogenic chlamydial species that are of significant concern for Australia.
It might also be of value to discuss or reference studies that examine the spread of animal chlamydia species in other countries and how they are been addressed (if they are), And whether importation into Australia is a major concern.
Author Response
Response to the reviewers
I thank the reviewers for their constructive comments to improve our manuscript. I have addressed each of the comments. A response for each of the reviewers’ points follows:
Reviewer 1
This review expertly summaries the current work and knowledge on the host distribution and epidemiology of Chlamydia species in regards to Australia. The review briefly summarises the current status of C. trachomatis and C. pneumoniae with a deeper dive on the current knowledge for C. psittaci and C. pecorum. I believe this review stands apart from other recent Chlamydia reviews by focusing on broad range of pathogenic chlamydial species that are of significant concern for Australia.
Author’s response: I would like to thank the reviewer for this feedback.
It might also be of value to discuss or reference studies that examine the spread of animal chlamydia species in other countries and how they are been addressed (if they are), And whether importation into Australia is a major concern.
Author’s response: Thank you for this suggestion, I have updated the review throughout with discussion or comparison to global studies.
See:
Section 1.1.1 and 1.1.2 added comparison to global trachoma and CT STIs distribution/elimination status.
Section 1.2 Cpn comparison to global knowledge.
Section 2.1 Psittacosis and livestock infection (Section 2.1.1) comparisons.
Section 3: koala chlamydiosis as a model disease.
Section 4: biosecurity discussion
Reviewer 2 Report
This is a review on Chlamydiae from Down Under: The curious cases of chlamydial infections in Australia written by M Jelocnik.
The title is promising and attractive. Also, I like the subheadings.
However, there are some concerns in the contents:
In many places, it is mentioned “we”, although there is only one author listed (MJ).
In addition, there are many inaccuracies and vague descriptions in the text.
This definitely has a focus on Australian research. Australian research has indeed been of high-quality with many prominent researchers (eg. P Timms, A Polkinghorne, W Huston just to mention a few currently active ones). In addition, Australia might be the only developed area with still trachoma present in certain areas (rural and remote though) and the animals in Australia seem to be a huge reservoir of various Chlamydiae. However, if this kind of review is to be published outside Australia, the research and findings have to be put in perspective and the world and research outside Australia cannot be excluded.
Maybe even to concentrate only on topic 2 (Chlamydial infections at the human-animal interface) and 3. (at the domesticated-wildlife animal interface) would be a fresh approach with a lot of Australian data (?).
Also, in many place it is mentioned studies in certain areas are lacking or can be improved. Those topics should not perhaps be discussed here, as the limited data available might mislead the readers (see eg. 1.2. C. pneumoniae chapter)
In the last section (5. Present challenges and future challenges) opinions of the leading researchers in the area are included as direct citations. Why were not these experts invited as (co)authors? This would have made the manuscript more balanced.
The abstract starts with a way too long sentence.
1.1.1. the subheading is not right
Trachoma and other ocular infection caused by C. trachomatis do have a different epidemiology
Only certain genotypes are associated with trachoma, but ocular infection can be caused by sexually transmitted genotypes (but they do not lead to blinding diseases).
Are the other causes for trachoma than ocular Ctr infections?
1.1.2. plural and singular in the same sentence
New Ct infections – are they ocular related to trachoma or STIs?
Myriad of Australian subpopulations (what about lesbian? Gay = MSM??, what are other?
LGV can be caused by other L types than L2b.
1.1.3. Do you believe that molecular studies can truly differentiate reinfection, persistent infection or treatment failure? That of course would be desirable.
Author Response
Response to the reviewers
I thank the reviewers for their constructive comments to improve our manuscript. I have addressed each of the comments. A response for each of the reviewers’ points follows:
Reviewer 2
This is a review on Chlamydiae from Down Under: The curious cases of chlamydial infections in Australia written by M Jelocnik. The title is promising and attractive. Also, I like the subheadings. However, there are some concerns in the contents:
In many places, it is mentioned “we”, although there is only one author listed (MJ).
Author’s response: I would like to thank the reviewer for the feedback and helpful suggestion for this review. We to I - was corrected as suggested throughout the text.
In addition, there are many inaccuracies and vague descriptions in the text.
This definitely has a focus on Australian research. Australian research has indeed been of high-quality with many prominent researchers (eg. P Timms, A Polkinghorne, W Huston just to mention a few currently active ones).
In addition, Australia might be the only developed area with still trachoma present in certain areas (rural and remote though) and the animals in Australia seem to be a huge reservoir of various Chlamydiae. However, if this kind of review is to be published outside Australia, the research and findings have to be put in perspective and the world and research outside Australia cannot be excluded.
Author’s response: Thank you for this very constructive suggestion, I have updated the review throughout with discussion or comparison to global studies. Although this review does have the focus on Australian research, many cases/situations can be used as a model globally. For example, I have added sentence in section 3.1: “As such, koala chlamydiosis also serves as a wildlife disease model, and many lessons from koala chlamydiosis can be transferred and utilised in other research.”
Maybe even to concentrate only on topic 2 (Chlamydial infections at the human-animal interface) and 3. (at the domesticated-wildlife animal interface) would be a fresh approach with a lot of Australian data (?).
Author’s response: The topic 2 and 3 are indeed very curious, with lot of recent developments, however the fact that Australia still remains developed and high-income country with endemic (hyperendemic even in some regions) trachoma; Ct as leading cause of STIs and perhaps under-recognised human Cpne infections is important to be communicated. I have taken onboard all your suggestion to improve this review (particularly 1.1 Ct and 1.2 Cpne chapter).
Also, in many place it is mentioned studies in certain areas are lacking or can be improved. Those topics should not perhaps be discussed here, as the limited data available might mislead the readers (see eg. 1.2. C. pneumoniae chapter)
Author’s response: Thank you for this very constructive suggestion, I have updated the review’s section on Cpne, this was certainly my omittance to expand on Cpne, and reworded to avoid misleading the readers.
In the last section (5. Present challenges and future challenges) opinions of the leading researchers in the area are included as direct citations. Why were not these experts invited as (co)authors? This would have made the manuscript more balanced.
Author’s response: All experts were fully aware of how this section was envisioned (this was also discussed with the editor) and were invited as co-authors. However, I have their full support, co-operation and utmost encouragement to lead this review as a single author and they to act as the interviewees. I am very grateful to their contributions and I have extended my gratitude in acknowledgments.
The abstract starts with a way too long sentence.
Author’s response: I have updated the abstract and split the sentence in three as per your helpful suggestion. The sentences now read: “In Australia, the most researched and perhaps the most successful chlamydial species are the human pathogen Chlamydia trachomatis, animal pathogen Chlamydia pecorum and zoonotic avian pathogen Chlamydia psittaci. C. trachomatis remains the leading cause of sexually transmitted infections in Australians and trachoma in Australian Indigenous populations. C. pecorum is globally recognised as the infamous koala and widespread livestock pathogen, whilst the avian C. psittaci is emerging as horse pathogen posing zoonotic risks to humans.”
1.1.1. the subheading is not right
Trachoma and other ocular infection caused by C. trachomatis do have a different epidemiology
Only certain genotypes are associated with trachoma, but ocular infection can be caused by sexually transmitted genotypes (but they do not lead to blinding diseases).
Are the other causes for trachoma than ocular Ctr infections?
Author’s response: I do agree with your comment, the start of the subheading: “Ocular Ct infections” can be misleading as other (UGT) than trachoma genotypes can cause Ctrac ocular infections/conjunctivitis. Hence the subheading is changed to: “Ct still a major cause of trachoma in a high-income developed country”. Furthermore, I have removed “(and Ct ocular infection) from another sentence in the same paragraph for clarity as this paragraph is focused on trachoma in Australia. By the WHO definition: Trachoma is a disease of the eye caused by infection with the bacterium Chlamydia trachomatis. I am not sure of other causes of trachoma.
1.1.2. plural and singular in the same sentence
Author’s response: I have updated and grammatically corrected the subheading: 1.1.2 Sexually transmitted Ct infections – STI infections with no intention of slowing down
New Ct infections – are they ocular related to trachoma or STIs?
Author’s response: This relates to STI Ct infections, and was corrected for clarity. Added “sexually transmitted” in the sentence:” In 2017, there were an estimated 255,228 new sexually transmitted Ct infections in Australians aged 15–29 years…”
Myriad of Australian subpopulations (what about lesbian? Gay = MSM??, what are other?
Author’s response: In this review Men who have sex with men (MSM) is referred to are male persons who engage in sexual activity with members of the same sex, regardless of how they identify themselves. Based on your suggestion, I have expanded this sentence to better describe sub-populations:” Sexually transmitted Ct infections affect a myriad of Australian sub-populations (e.g. lesbian, gay, bisexual, heterosexual, men who have sex with men (MSM) and other (pansexual, transgender, queer/questioning, intersex)) and …”
LGV can be caused by other L types than L2b.
Author’s response: I have removed this based on your suggestion.
1.1.3. Do you believe that molecular studies can truly differentiate reinfection, persistent infection or treatment failure? That of course would be desirable.
Author’s response: To some extent and using (standardised) high resolution molecular tools, I believe that it is possible to dissect the strains. However, as variety of schemes are used presently, and based on your suggestion, I have reworded this sentence: “Molecular Ct studies can certainly be improved in Australia as they can be utilised as a valuable epidemiological tools to track new and/or persistent infections or determine treatment failure, particularly as there is growing concern about the latter”.
Round 2
Reviewer 2 Report
The manuscript has improved considerably.
Still I cannot stop worrying the format in the last section.
Didn't the Australian researcher with international reputation want to be coauthors (what does full support truly mean?)? I wonder if there is a need to publish a technical single-author paper for some reason (advancing ones career or similar) in Australia? Knowing this would make me more understanding merciful.
But otherwise this manuscript (except the section 5) reviews thoroughly the Australian Chlamydia scene.